# How Do Neural Networks Overcome Label Noise?

**Amnon Drory, Shai Avidan & Raja Giryes**
School Of Electrical Engineering
Tel-Aviv University
`amnondrory@mail.tau.ac.il, avidan@eng.tau.ac.il, raja@tauex.tau.ac.il`

## Abstract

This work provides an analytical expression for the effect of label noise on the performance of deep neural networks.

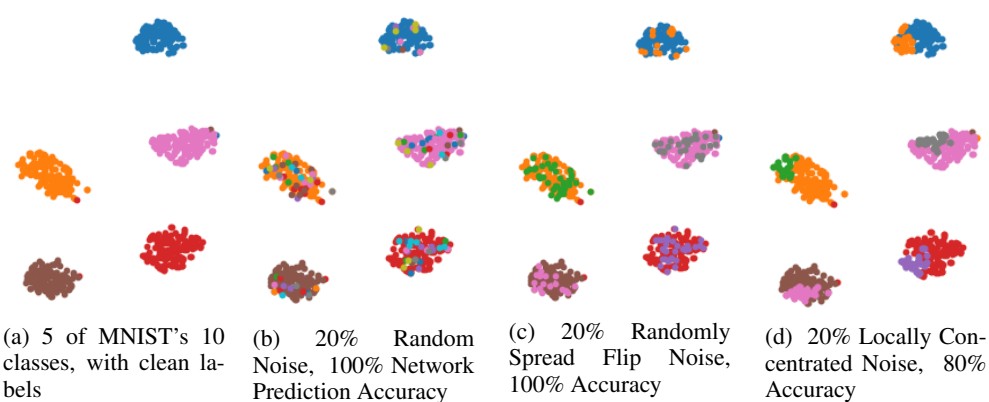

(a) 5 of MNIST's 10 classes, with clean labels

(b) 20% Random Noise, 100% Network Prediction Accuracy

(c) 20% Randomly Spread Flip Noise, 100% Accuracy

(d) 20% Locally Concentrated Noise, 80% Accuracy

Figure 1: Different types of random label noise. DNNs are extremely resistant to noise when the corrupted labels are randomly spread in the training set (b,c). However, they are not resistant at all to noise that is locally concentrated (d, produced using $k$-means with k=5). Features are all in 256-dimensional space, and reduced to 2D for visualization using tSNE (van der Maaten & Hinton (2008))

## 1 Introduction

It has been demonstrated in several works (e.g. Krause et al. (2015)) that Neural Networks (DNNs) are extremely robust to label-noise. In some experiments, DNNs are shown to produce high accuracy results even when trained on a dataset where the wrong labels outweigh the correct labels (e.g., Rolnick et al. (2017)). In this work, we offer an explanation for this phenomenon: similarly to the K-Nearest Neighbors algorithm (KNN), DNNs use an entire neighborhood of training samples to predict the label of a test sample. We show analytically that as long as the noisy labels are *well spread* in the feature space, accuracy remains high even with high levels of label noise. Our main contribution is providing an analytical expression for the expected accuracy of the network at any given noise level. We show that this expression fits with experimental results quite well, and allows us to estimate the size of the neighborhood.

## 2 Label Noise

In most works that examine DNNs' resistance to label noise, the network's goal is to perform classification, i.e. to assign a label $y \in \mathcal{L} = \{\ell_1, \ell_2, \ldots, \ell_L\}$ to a test sample $x$ (typically an image). The network is trained on a noisy dataset $\{\tilde{x}_i, \tilde{y}_i\}_{i=1}^{N}$, which is derived from a clean dataset $\{x_i, y_i\}_{i=1}^{N}$ by changing some of the labels. The simplest scenario is where a random subset of the training samples receive a random new label, uniformly sampled from $\mathcal{L}$. We define the *noise level*, $\gamma$, as the

fraction of the training set that get their labels re-assigned, and we say that these samples have been *corrupted*. This setting is used, for example, by Bekker et al. (Bekker & Goldberger (2016)) and we will refer to it as *random label-noise*.

The second common setting considered is *flip label-noise*: Each label $\ell_i$ has one counterpart $\ell_j$ with which it may be replaced. Again, $\gamma \cdot N$ samples are randomly selected, and for each one the true label is replaced with its counterpart. This setting is used, for example, by Reed et al. (Reed et al. (2014)). A more general setting, which includes the previous two types as a special case, is *confusion-matrix label-noise*. In this setting, the probability of the new label depends on the original label, and is described by a conditional probability function (Sukhbaatar & Fergus, 2014).

In all of the settings we described, the samples to be corrupted are randomly selected. As a result, the noisy labels are *randomly spread* in the feature space. In contrast to that, in the *locally concentrated noise* setting, the noisy labels are locally concentrated in some feature space. To generate this type of noise we first train a network on a clean dataset, and then consider the outputs of the penultimate layer. For each class separately, we perform $k$-means in this space, and flip the labels of one cluster per class. Each class $\ell_i$ has one alternative class $\ell_j$ to which the noisy labels are flipped. $k$-means with different values of k result in different noise-levels, from roughly 10% when k=10, to roughly 50% when k=2.

## 3 THEORETICAL ANALYSIS

Fig. 2 demonstrates that while the accuracy of a neural network does not change in the presence of randomly spread label noise, it decreases significantly if the noise is concentrated in one location. We argue that this is due to the fact that a network tends to encapsulate the local distribution of training samples in the vicinity of each sample. Using this assumption, we derive hereafter a bound that predicts the error of a network as a function of the fraction of noisy labels $\gamma$.

When the noisy labels are concentrated in the feature space, instead of spread randomly, then we can expect mostly two types of neighborhoods to exist: *clean neighborhoods* which will result in correct classification, and *completely corrupted neighborhoods* where all training samples have incorrect labels, which will lead to incorrect prediction. Assuming the test samples are uniformly spread in the feature space, the conclusion would be that

For the settings where noise is randomly spread, and with the help of a few simplifications, A mathematical expression for the accuracy given the noise level can be produced. In experiments, we show that empirical accuracy vs. noise level fit with the curves produced by this formula quite well. In developing an analytical formula we follow this intuition but make a stronger assumption: that the network perform a K Nearest Neighbors algorithm. This means that the neighborhood that determines the output label is always a set of K samples, regardless of which test sample we are looking at, and regardless of noise level. Another assumption is that in the clean dataset, all samples in the K-neighborhood have the same label, $\ell_{correct}$, which is also the correct label for the test sample. This simple assumption would mean that when the network is trained on the clean dataset, its prediction should always be correct, and therefore its accuracy will be 100%. If a network is trained with *noisy* labels, however, the probability of a wrong prediction is the probability that the plurality label in the K-neighborhood *is different* from $\ell_{correct}$. We call this situation a *plurality label switch*.

The expected overall accuracy of the network, $A$, is defined as the expected fraction of test-set samples that will be classified correctly:

$$A = \sum_{\ell \in \mathcal{L}} f_\ell \cdot Q(\ell, K, P, \gamma) \tag{1}$$

where $f_\ell$ is the fraction of test samples whose true label is $\ell$. $Q(\ell, K, P, \gamma)$ is the probability of a test sample being correctly classified with its correct label $\ell$. An expression for $Q$ can be derived by going over all possible assignments of noisy labels to the K-neighborhood, and counting those where the plurality-label is the same as the correct label:

$$Q(\ell, K, P, \gamma) = \sum_{n_1} \sum_{n_2} \cdots \sum_{n_L} \binom{K}{n_1, n_2, \ldots, n_L} q_1^{n_1} \cdot q_2^{n_2} \cdot \cdots \cdot q_L^{n_L}, \tag{2}$$

where $q_i = q_i(\gamma, P)$ is the probability for any sample in the K-neighborhood to be labeled $\ell_i$, $n_i$ is the number of appearances of the label $\ell_i$ in the K-neighborhood, $P(\tilde{y}|y)$ is the alternative label probability matrix (the confusion-matrix), and the summation is only over assignments where $\ell$ is the plurality label.

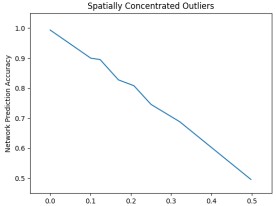 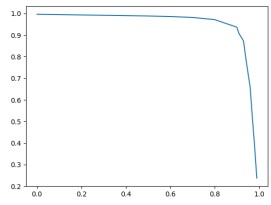 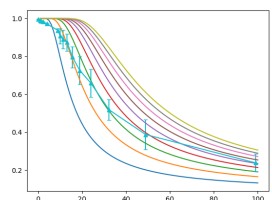

(a) Concentrated Noise, Experimental accuracy as function of $\gamma$ (fraction of corrupted labels)

(b) Random Noise, Experimental accuracy as function of $\gamma$ (fraction of corrupted labels)

(c) Experimental curve overlayed over analytical curves

Figure 2: Experimental results. (a) Concentrated noise (b) Random Noise (c) Comparison of Experimental and Analytical curves showing the accuracy vs. noise level. The x axis is $\frac{\gamma}{1-\gamma}$ to better focus on areas where most degradation occurs. Analytical curves are shown for K=100,200,...,1000 (from bottom to top). The experimental curve shows the mean accuracy (over 10 experiments) and the standard deviation.

## 4 EXPERIMENTS

We produce empirical accuracy vs. noise level curves for the *random* and *concentrated* noise models described in Section 2, by training NNs using training sets with different levels of noise, then testing the accuracy on the (clean) test set. We use the MNIST dataset, and our network is based on Lenet-5, with the following changes: (i) Each convolution layer is duplicated, resulting in 4 conv layers instead of 2; (ii) Zero-padding is added to all convolutions; and (iii) Batch Normalization (Ioffe & Szegedy (2015)) is added after every convolution layers. For the *concentrated* setting, we use $k$-means with $k$ values of 10, 8, 6, 4, 3, and 2. For the *random* setting, we train 10 networks for each of a range of noise levels $\gamma$, and report the mean accuracy, as well as its standard deviation. In parallel, we produce theoretical curves for a range of values of $K$ using a multi-threaded C++ implementation of (2). Figure 2 presents the results. In Fig 2(a) we see that as expected, the neural network is not resistant to locally concentrated label-noise, and the accuracy drops by the same amount that the noise increases. In figure 2(c) we see that the empirical curve for the *random* setting fits reasonably well with the analytical curves.

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
