# OpenReview forum: "How Do Neural Networks  Overcome Label Noise?"
_ICLR.cc/2018/Workshop — Reject_

### Official Review · AnonReviewer1 · 2018-03-09
**Filtering Adversarial Examples**

**Rating:** 4
**Confidence:** 5

**Review:**

This work proposes an analytical expression for the effect of label noise on the performance of deep neural networks. The main issue is that it does not provide such an expression. On a single example, the work illustrates that with random label noise the model's classification accuracy degrades more gracefully than with concentrated noise. This is obvious and falls short of the promise.

---

> ### Author Response · Authors · 2018-04-12
> **Analytical expression for the resistance of networks to random noise**
>
> Indeed, in three pages abstract, we are limited in the amount of technical details we can give.
> Full details of our analytical expression with its derivation is provided in the full paper:
> https://arxiv.org/abs/1803.11410

---

### Official Review · AnonReviewer3 · 2018-03-10
**An interesting work with relatively weak validation**

**Rating:** 6
**Confidence:** 3

**Review:**

This work provides an analytical expression for the effect of label noise on the performance of deep neural networks.

The studied problem is interesting. But the proposed method can be better motivated. The validation is relatively weak.

---

> ### Author Response · Authors · 2018-04-12
> **More validation**
>
> Indeed, in three pages abstract, we are limited in the amount of explanation and validation we can give.
> Full details of the derivation of the analytical expressions for predicting the resistance of the network to label noise with more experimental validation is provided in the following paper:
> https://arxiv.org/abs/1803.11410

---

### Decision · Program_Chairs · 2018-03-20
**ICLR 2018 Workshop Acceptance Decision**

**Decision:**

Reject

**Comment:**

Based on the reviews, this paper has not been accepted for presentation at the ICLR workshop. However, the conversation and updates can continue to appear here on OpenReview.